# Oropouche fever outbreak in Brazil: Key factors behind the largest epidemic in history

Camila Lorenz[1*], Thiago Salomão de Azevedo[2,3], Maria Anice Mureb Sallum[4], Francisco Chiaravalloti-Neto[4]

1 Department of Parasitology, Instituto Butantan, São Paulo, Sao Paulo, Brazil, 2 Secretary of Health, Municipality of Santa Barbara d'Oeste, Sao Paulo, Brazil, 3 Department of Biodiversity, Institute of Biosciences, UNESP, Rio Claro, Sao Paulo, Brazil, 4 Department of Epidemiology, School of Public Health, University of Sao Paulo, São Paulo, Sao Paulo, Brazil

☉ These authors contributed equally to this work.
* camilalorenz@usp.br

## Abstract

Oropouche virus (OROV) is an arthropod-borne virus responsible for outbreaks of Oropouche fever (ORO) in Central and South America since the 1950s. Herein, we investigated the climatic and socioenvironmental factors contributing to the reemergence of ORO in Brazil in 2024, culminating in the largest epidemic in the country's history. Accordingly, we conducted a modeling study to identify areas with the highest incidence of OROV in Brazil based on confirmed human cases between the 2020 and 2024 outbreaks and socioenvironmental variables. Our analysis utilized Maxent software, a machine learning algorithm for species distribution modeling, along with SatScan software to identify high- and low-risk spatial clusters. A total of 8,258 ORO cases were serologically confirmed in Brazil in 2024 and 108 in 2020/2021. The distribution of OROV differed markedly in 2020 and 2024: in 2020, cases were primarily confined to the Amazon region, while in 2024, cases expanded across nearly the entire country. High-risk areas showed higher temperatures and precipitation, and land-cover and land-use change (LCLUC) appeared to be key factors in ORO distribution. Upon comparing deforestation rates between 2020 and 2023, we noted a sharper increase in the expansion of pasture cover and soybean plantations in high-risk regions. Moreover, municipalities in high-risk clusters tended to face greater socioeconomic challenges, including poverty and restricted access to healthcare. Our study identified areas vulnerable to OROV circulation, providing valuable insights to support the establishment of robust public health policies that must be prioritized and strengthened in the context of climate change.

## Introduction

With the rise in global connectivity, the Americas have experienced the emergence and reemergence of several arboviruses over the past 50 years, including Zika,

**Data availability statement:** All relevant data are publicly available without restriction from the Brazilian Ministry of Health at the following URL: https://www.gov.br/saude/pt-br/assuntos/saude-de-a-a-z/o/oropouche/painel-epidemiologico.

**Funding:** C.L. was supported by São Paulo Research Foundation (FAPESP) grant number 2022/13367-9. The funder had no involvement in the study design, data collection and analysis, decision to publish, or manuscript preparation.

**Competing interests:** The authors have declared that no competing interests exist.

dengue, West Nile, and chikungunya [1]. Although poorly understood, Oropouche fever (ORO) is a notable, emerging zoonotic disease, which is caused by the Oropouche virus (OROV) and was first identified in a forest worker with fever from Vega de Oropouche in Trinidad and Tobago in 1955 [1]. OROV, a member of the genus Orthobunyavirus, is characterized by a negative-sense, single-stranded RNA structure and a spherical lipid envelope genome [2]. OROV comprises three single-stranded RNA segments and is enveloped by a helical nucleocapsid that encodes its essential components. However, the details of the OROV replication cycle remain unknown.

ORO is a syndrome that mimics other vector-borne diseases, such as dengue, Zika, and Mayaro fever, and is associated with common symptoms such as headache, muscle pain, and fever [3,4]. At least four genotypes of ORO have been identified during outbreaks in Trinidad and Tobago, Peru, Panama, and Brazil [5]. OROV persists in a sylvatic-rural cycle with wildlife hosts and arthropod vectors. Although research is ongoing, nonhuman primates, some wild bird species, and pale-throated sloths may act as hosts [6]. OROV transmission from wildlife to humans occurs mainly through the midge *Culicoides paraensis*. In addition, *Culex quinquefasciatus*, a medium-sized mosquito common in tropical and subtropical areas globally, may play a role [7,8]. In temperate regions, *Culex pipiens* may contribute to transmission, with hybrid zones located in the transition zone between the subtropical and temperate regions.

From the beginning of 2024 through early December, 13,014 confirmed cases of ORO were documented in 12 countries in the Americas, with Brazil accounting for the highest proportion (10,940 cases, including two deaths). Other affected countries included Bolivia, Colombia, Cuba, Ecuador, Guyana, Panama, and Peru. Additionally, travel-related imported cases were reported in the USA (90 cases), Canada (2 cases), and Europe (30 cases) [9]. In 2024, the virus was detected in urban/rural areas where no transmission had previously been reported. Infection-related deaths were documented, along with cases of vertical transmission (from mother to baby during pregnancy), including instances of fetal death and congenital anomalies (PAHO, 2024).

Land-cover and land-use change (LCLUC) and climatic factors have been suggested as drivers of disease emergence, including ORO [3]. Accordingly, investigating environmental conditions that facilitate infectious diseases could help develop strategies for preventing disease emergence. However, there is a considerable lack of knowledge regarding the epidemiology, ecology, and pathogenesis of OROV. Addressing these gaps is essential for developing risk assessments and effective public health strategies to control this arbovirus [1]. Given the epidemic potential of OROV, limited data in the literature, and recent spikes in outbreaks beyond endemic areas [10], identifying regions at risk of OROV spread to human populations is crucial. In the current study, we estimated the geographic potential distribution of OROV in Brazil to determine the likelihood of OROV occurrence based on environmental conditions during the current outbreak period (2024) and compared it with the 2020/2021 scenario.

## Materials and methods

The methodology outlined below is based on the Lorenz et al. [11] approach, who analyzed the distribution and modeling of West Nile virus in South America. All records of ORO in Brazil reported by Ministry of Health [12] between 2020 and 2024 were analyzed, confirmed by molecular biology or serology. The criterion for inclusion of a municipality in the analysis was presence ≥ 5 of confirmed ORO human cases, as a proxy of autochthonous transmission. The points with presence data are the coordinates of the centroid of the municipality in which the case was registered.

To verify the ecological and climatic variables related to the presence of OROV, we explored the associations between the locations of the records and the variables: 1) mean temperature range (diurnal and annual), 2) annual mean temperature, 3) isothermality, 4) temperature seasonality, 5) annual precipitation and 6) precipitation seasonality. These parameters were elected considering their potential to shape the life cycles and biological conditions of vectors, humans, and non-human hosts, or their association with more immediate causal factors. WorldClim – Global Climate Data database [13] was the reference for all climatic data, including representative observational data spanning from 1950 to 2000, interpolated to a resolution of 30 arc seconds (approximately 1 km). Climate data for the most recent years, 2000–2023, was obtained from the National Institute for Space Research (INPE) [14].

Environmental variables for OROV modeling were preselected through principal component analysis (PCA). After PCA, we identified the four most representative variables, which were then utilized for modeling analysis. The maximum entropy (MaxEnt) distribution model was employed, surpassing other species distribution models [15,16]. This model requires data about the presence of the species being modeled, along with its variables related to the climate and environment (categorical). A replicate run was conducted as cross-validation, with 10000 iterations fixed. The model underwent 30 runs, each with a 10% difference in localities to estimate parameters and their precision. Occurrence points and similarity measures of environmental variables were interpolated to create potential distribution maps for each pixel. For further analysis, the final predictions were exported to ArcGIS 9.1.

Susceptibility areas were identified using the Threshold (Susceptibility Limit) feature in Maxent software. This procedure was performed to evaluate the suitability of ORO transmission areas and to pinpoint locations where the probability of this arbovirus has expanded. The selected model was the 10 Percentile Training Presence, which establishes the threshold at the probability value where 90% of training occurrences are predicted to occur above the threshold, leaving only 10% below it. This threshold helps to avoid overfitting and focuses on the most suitable areas [15,17]. This procedure produced two binary maps showing the susceptible areas for ORO occurrence for the periods of 2020/2021 and 2024. A cartogram was created through a Boolean map analysis in ArcGIS to show areas where the disease has retracted, remained stable, or expanded.

We performed spatial clustering analyses of ORO incidence using SaTScan v10.0 software [18]. Retrospective spatial scan statistics helped identify regions with higher-than-expected ORO incidence rates (hotspots/clusters). A Poisson probability model [19] was applied to detect high incidence rates under the following conditions: clusters were limited to 6% of the population and were assumed to have a circular shape. The Gini index option in SaTScan for purely spatial analysis [20] was used to determine the maximum population size for the scanning window. To increase statistical power and ensure finite p-values, we set the number of Monte Carlo replications to 999 [18]. Clusters with p-values less than 0.05 were deemed statistically significant. Demographic information per municipality were acquired from the Brazilian Institute of Geography and Statistics [21]. Data on deforestation and LCLUC were obtained through the MapBiomas platform [22].

Once the higher- and lower-risk clusters for ORO were identified through spatial analysis, the demographic and environmental variables of the municipalities in each group were compared (see S1 Dataset). An Excel spreadsheet was created, where the values for the higher-risk municipalities were placed in the first column and the lower-risk municipalities in the second. For each demographic and environmental variable, we used a t-test to compare the means between both groups, assuming equal variances. The null hypothesis stated that the means of the same variable were equal, and the significance level was set at 5%.

## Results

A total of 8,258 laboratory-confirmed cases of ORO were reported in Brazil in 2024 and 108 in 2020/2021. Among these, 6,735 cases were considered, encompassing 170 municipalities. A noticeable shift was observed between the distribution of OROV in 2020/2021 and in 2024 (Fig 1). In 2020, cases were primarily confined to the Amazon region, while in 2024 there was an expansion across nearly the entire country (Fig 2). The main climatic factors affecting the virus's potential distribution in 2020 were related to annual precipitation and temperature (annual mean temperature and temperature range, and diurnal temperature range). In 2024, however, the factors most strongly associated with OROV presence were exclusively temperature (annual mean temperature, mean diurnal temperature range, and temperature seasonality) (Fig 1).

Regarding spatial scan analysis of the ORO cases that occurred in 2024 and confirmed in laboratory, four statistically significant high-risk spatial clusters and three low-risk clusters were identified (Fig 3). The highest-risk areas during the analyzed period were predominantly found in the Amazon region (RR = 98). The other three secondary high-risk clusters were primarily located in municipalities along the Brazilian coast, particularly in the state of Espírito Santo (RR = 15.9 and 14.8) and Bahia (RR = 4.0).

The statistical analysis of demographic and environmental variables between municipalities in the high-risk spatial clusters and those in the low-risk clusters revealed significant differences between both groups. Significant p-values were obtained from all analyses (Table 1). Higher-risk areas had higher temperatures, greater precipitation, and a higher percentage of forest cover. LCLUC appears to be a key factor in ORO distribution. Moreover, municipalities located in high-risk clusters tend to be poorer and have limited access to healthcare. While these areas generally have more forest cover and less pastureland compared to low-risk regions, a comparison of deforestation rates between 2020 and 2023 reveals a raising expansion of pasture cover and soybean plantations (Table 1 and S1 Fig).

## Discussion

Our niche-modeling analyses revealed the geographical expansion of ORO into previously non-endemic areas, including densely populated regions in Brazil, such as Bahia, Espírito Santo, and Rio de Janeiro, where increased human-vector interactions could amplify the risk of transmission. This spread highlights the growing threat posed to the large immunologically naïve population across the Americas, coupled with the widespread presence of *C. paraensis* and other potential urban OROV vectors, including *C. quinquefasciatus* and *C. pipiens*, extending from the southeastern United States to Uruguay [23]. These ubiquitous house mosquitoes may participate in urban transmission, although vector competency evaluations have yielded mixed results [7,8]. Limited data exist regarding specific vectors associated with recent urban outbreaks, although viral RNA has historically been detected in biting midges, including *C. paraensis*. According to Pinheiro et al. [24], a single infected *Culicoides* midge can transmit OROV. Moreover, *Culicoides* midges are a global public health concern, given that they function as known vectors for several other arboviruses, including equine encephalitis and Schmallenberg virus [25]. Approximately 96% of these midge species feed on the blood of humans and wild mammals [25].

Scachetti et al. [26] reported that the 2023/2024 outbreaks were driven by a novel OROV reassortant, which exhibited replication at approximately 100 times higher titers in mammalian cells than in the prototype strain. Although the implications of this faster replication of OROV transmission by vectors remain uncertain, it is plausible that increased viral fitness could result in higher viremia levels, thus enhancing the efficiency of vector infection during blood feeding [27]. Serum samples collected in 2016 from individuals previously infected with OROV revealed a statistically significant reduction (at least 32-fold) in the neutralizing capacity against the reassortant strain compared with the original prototype [26]. Moreover, our analysis predicted a higher incidence of ORO in Northern Brazil in 2024, a region with documented OROV circulation dating back to the 1950s [28]. This suggests that human populations in these historically endemic areas remain exposed to OROV and may be susceptible to reinfection owing to diminished neutralizing antibody responses against emerging variants, such as the 2023/2024 reassortant strains [26]. This trend is particularly concerning in highly suitable

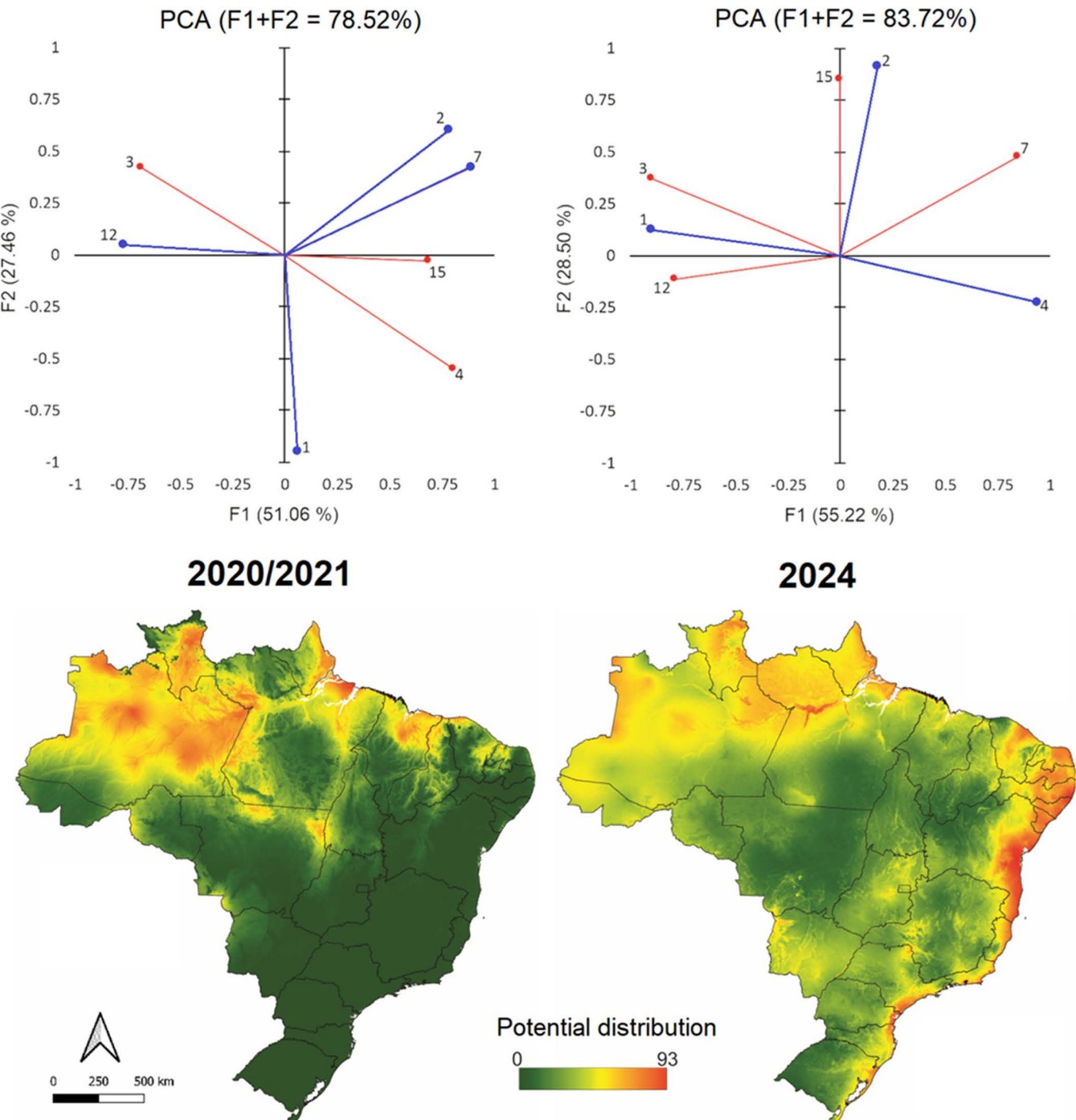

**Fig 1. Predicted ecological suitability for OROV transmission across Brazil based on ecological niche modelling.** The maps show the potential distribution in 2020/2021 and 2024. PCA shows the most representative environmental variables for the OROV model: (1) Annual mean temperature; (2) Mean temperature diurnal range; (3) Isothermality; (4) Temperature seasonality; (7) Temperature annual range; (12) Annual precipitation; (15) Precipitation seasonality. We chose the four most representative variables (blue), which were employed for Maxent analysis. The maps were built using ArcGIS version 9.1. Source of map base layers: IBGE: https://www.ibge.gov.br/geociencias/cartas-e-mapas/mapas-estaduais.html. Open-source CC BY 4.0 license.

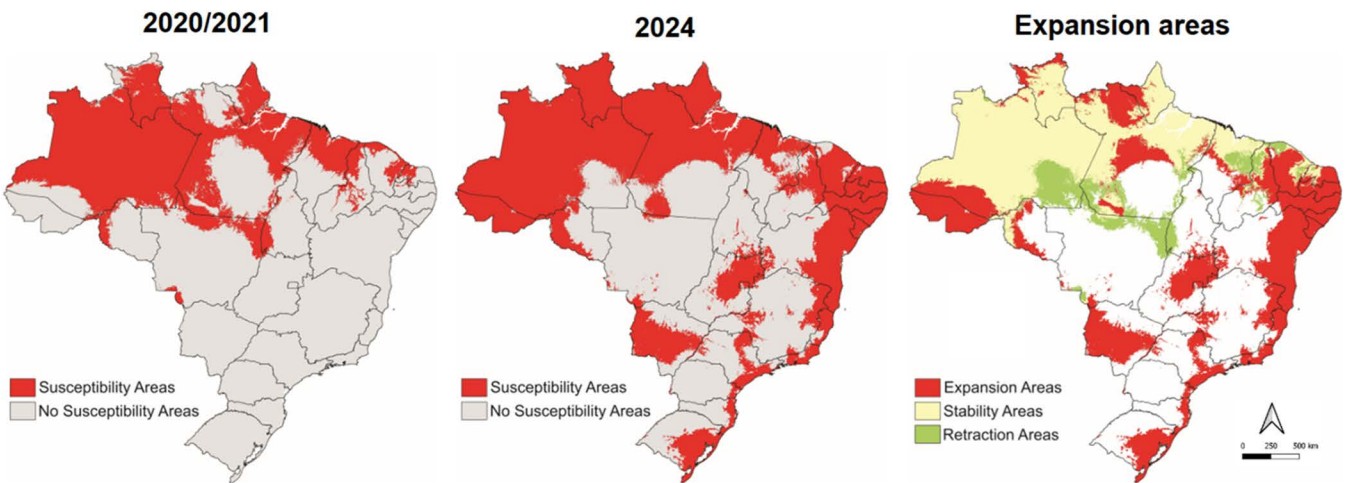

**Fig 2. Susceptibility occurrence maps of OROV in two scenarios (2020/2021 and 2024) and expansion areas based on ecological niche model-ling.** Data obtained from Maxent software version 3.3.3. The maps were built using ArcGIS version 9.1. Source of map base layers: IBGE: https://www.ibge.gov.br/geociencias/cartas-e-mapas/mapas-estaduais.html. Open-source CC BY 4.0 license.

areas along the coast of Brazil, home to more than half of the 200 million inhabitants [29]. Such a scenario could facilitate the entry and establishment of OROV throughout the Americas and beyond, similar to the previous spread of dengue, Zika virus, and chikungunya virus [26,30].

The Amazon remains the primary endemic area for ORO, and various factors create an environment conducive to its persistence. The humid and warm climate of the region, coupled with dense vegetation and frequent rainfall, provides ideal conditions for the proliferation of *C. paraensis* midges. However, the expansion of human activities, including defor-estation and urbanization, has altered the natural habitats of these vectors, bringing them into close contact with human populations and increasing transmission risks [10]. Furthermore, an increase in ORO cases in non-Amazonian states has been linked to increased human mobility and climate change. For example, urbanization in areas lacking proper infra-structure may intensify the rapid expansion of vector habitats, leading to the emergence of new urban hotspots for trans-mission. Additionally, the potential novel reassortment of OROV could enhance its adaptability to new vectors or increase its virulence, further enabling its spread to previously unaffected regions [26]. As its symptoms are similar to those of other arboviruses, such as dengue, ORO is often underreported, complicating the diagnosis. Two deaths related to ORO were confirmed in Bahia, Brazil, on July 25, 2024 [9]. Recent findings from Pernambuco and Acre indicate cases of vertical transmission, highlighting epidemiological challenges that mirror those experienced during the 2015/2016 Zika virus epi-demic [10].

Herein, we identified that higher-risk areas were mainly concentrated in the Amazon and coastal regions of Bahia and Espírito Santo States during the 2024 epidemic. Martins-Filho et al. [10] and Tegally et al. [29] also showed that these regions are hotspots of medical concern for ORO cases. All higher-risk areas were consistent with higher temperatures, precipitation, and a higher percentage of natural vegetation, indicating a possible association between these factors and ORO incidence. The average temperature in these areas was ~27°C, which may provide an ideal setting for OROV replication within its vector, as indicated by initial studies on the thermal biology of biting midges [31]. When examining land-use patterns, high-risk zones generally feature more extensive forest cover and less pastureland than low-risk zones. However, a comparison of deforestation rates between 2020 and 2023 revealed a substantial expansion in pasturelands and soybean plantations. These changes in land use could profoundly influence mosquito populations, affecting factors such as oviposition, population density, and host-seeking behavior. Multiple studies have demonstrated that shifts in

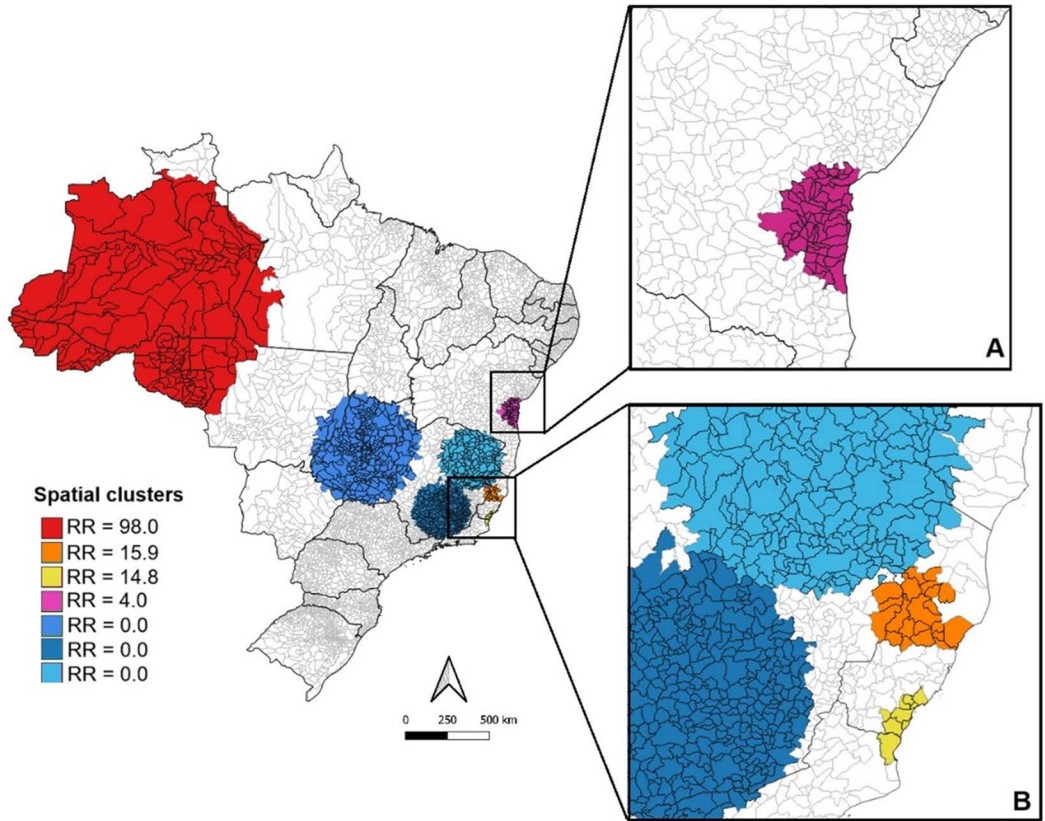

**Fig 3. Spatial analysis clusters with high and low incidences of ORO according to relative risk (RR) during 2024 period.** (A) High-risk cluster in Bahia State (B) High-risk clusters in Espírito Santo State. Data obtained from SatScan software version 10.0. The maps were built using ArcGIS version 9.1. Source of map base layers: IBGE: https://www.ibge.gov.br/geociencias/cartas-e-mapas/mapas-estaduais.html. Open-source CC BY 4.0 license.

LCLUC often result in the loss of mosquito habitats, host species, and natural predators, which, in turn, can alter vector behavior and dynamics. Such environmental modifications could force mosquitoes to search for new blood sources and alternative habitats, potentially resulting in increased host contact rates, promoting the emergence of disease spillovers, and introducing new infections to vulnerable human populations [32].

Landscape disturbances are potential driving forces of ORO outbreaks [3]. For instance, highway construction was considered a potential factor in the ORO outbreak in Belem, Brazil, in 1962 [33]. Likewise, landscape modifications have also been implicated in the spread of ORO, such as the 2010 outbreak in the San Martin Department of Peru [34] and the 2011 outbreak in Cutervo Province, Peru [35].

Compelling evidence indicates preferential circulation of certain OROV lineages in areas linked to cocoa and banana plantations, as highlighted by Tegally et al. [29]. The high-risk clusters were primarily located in regions with considerable cocoa production, particularly in the coastal areas of Bahia and Espírito Santo. These findings are consistent with existing evidence, revealing that *C. paraensis* larvae thrive in microhabitats of decaying organic matter and debris from banana and cacao plantations [36,37]. Such insights are valuable for informing public health strategies and interventions, particularly vector control efforts around banana and cocoa plantations, especially those in proximity to urban centers. A recent study has underscored the importance of banana and cocoa crops in areas experiencing ORO epidemics [38].

**Table 1. Comparison of means between high and low risk clusters of ORO incidence in 2024 (t-test, two samples assuming different variance). The number in parentheses indicates the standard deviation. *Per 10,000 inhabitants.**

| Variable | High-risk cluster | Low-risk cluster | *p* value |
|---|---|---|---|
| *Climatic* | | | |
| Mean annual precipitation (mm) | 151 (49) | 117.6 (21.4) | 0.01 |
| Minimum temperature (ºC) | 19.9 (2.3) | 15.6 (2.1) | 0.001 |
| Maximum temperature (ºC) | 33.5 (2.4) | 28.1 (2) | 0.01 |
| Mean temperature (ºC) | 26.7 (2.1) | 21.9 (2) | 0.01 |
| *Socioeconomic* | | | |
| PIB per capita | 25,975.2 (4120.4) | 35,224.8 (5481.5) | 0.001 |
| Incidence rate of infectious/parasitic diseases* | 62.2 (8.3) | 43.1 (4.2) | 0.001 |
| Number of basic health units* | 7.1 (3) | 9.5 (3.9) | 0.001 |
| Number of SUS doctors* | 8 (5.4) | 11.3 (8.1) | 0.001 |
| Primary care coverage (%) | 90.1 (13.3) | 94.2 (11.9) | 0.001 |
| *LCLUC* | | | |
| Forest coverage (%) | 51.2 (26.7) | 19.2 (12.3) | 0.01 |
| Pasture coverage (%) | 28.4 (26) | 43.5 (19.3) | 0.01 |
| Urbanization (%) | 78.9 (10.2) | 91.5 (14.3) | 0.001 |
| Difference between pasture cover (2020–2023) (%) | 16.9 (13.7) | −3.1 (7.9) | 0.01 |
| Difference between soybean cover (2020–2023) (%) | 76.4 (28.2) | 21.9 (10.4) | 0.01 |

Our study highlights the considerable role of environmental factors in influencing OROV transmission dynamics. However, climatic conditions alone cannot account for the numerous disease outbreaks, such as the Zika epidemic [39] and the long-standing endemics of malaria and dengue [40]. Susceptibility to these diseases is shaped by the complex interaction between climatic factors and structural elements, including infrastructure quality and educational level [41]. This dynamic is particularly relevant in countries such as Brazil, which continue to face low levels of overall sanitation coverage and challenging socioeconomic conditions, particularly in the Amazon region [42]. Notably, our analysis revealed that municipalities within high-risk clusters were often characterized by higher poverty rates and limited access to healthcare, with poverty being intrinsically connected to the prevalence of neglected diseases [43]. Furthermore, the Amazon and northeastern regions of Brazil exhibit the highest levels of climate vulnerability [41]. In Brazil, inadequate living conditions and limited access to healthcare services have substantially contributed to the ongoing prevalence of this disease. Several urban areas, particularly low-income neighborhoods, face major challenges in terms of sanitation infrastructure. Although Brazil provides universal health coverage through its public healthcare system, the Sistema Único de Saúde, access remains uneven, especially in rural regions. In the northeastern part of the country, half of the rural population reportedly lives more than 5 km from the nearest healthcare facility, while 60% live over 10 km away [44]. Furthermore, the distribution of medical professionals is heavily skewed toward urban centers, leaving rural areas underserved. These disparities act as barriers to timely medical care, which, in turn, contribute to the rising number of OROV cases in rural communities, as many residents are discouraged from seeking treatment.

The potential for multiple events of OROV introduction into areas beyond the Amazon region poses a considerable risk of widespread transmission, particularly among immunologically naïve populations or in areas where emerging viral variants may evade preexisting immunity [26]. Furthermore, several high-risk regions, especially in northeastern and central-western Brazil, remain underrepresented in surveillance efforts, which may result in undetected viral transmission owing to insufficient sampling. Accordingly, prioritizing these poorly surveyed areas for targeted surveillance is crucial for early viral detection and prompt intervention to mitigate future outbreaks [29]. The methodologies employed in the current study offer valuable tools for addressing these surveillance gaps, providing a basis for spatial prioritization and a deeper

understanding of ORO epidemiology. With the adaptation of OROV to urban settings, the incidence of cases is likely to increase, particularly due to climate change. Therefore, implementing comprehensive educational initiatives focusing on ORO prevention and treatment for both communities and healthcare professionals is a pivotal public health measure for curbing the growing number of cases.

A comprehensive approach must be established to effectively address OROV epidemics in Brazil and other South American countries. This involves prioritizing research to identify zoonotic reservoirs, assess human-to-human transmission potential, and understand viral dynamics through long-term studies and interdisciplinary collaborations. Leveraging geographic information systems and environmental modeling can help identify high-risk areas and enable targeted public health interventions. Integrating human, animal, and environmental health data within a one-health approach can enhance surveillance, predictive modeling, and cross-sectoral response strategies.

## Conclusions

In summary, our findings provide essential insights into the possible factors involved in OROV transmission and serve as data for future public health strategies and research focused on preventing and mitigating outbreaks. High-risk areas are typically characterized by elevated temperatures, increased rainfall, and other environmental conditions that promote the spread of the virus. In addition, municipalities in high-risk clusters tend to face greater socioeconomic challenges, including poverty and limited access to healthcare. LCLUC appears to play a crucial role in ORO distribution. Despite having more forest cover and less pastureland than the low-risk areas, these regions experienced a more pronounced expansion of pastureland and soybean plantations between 2020 and 2023. These findings demand enhanced public health preparedness, with an emphasis on creating vaccines that can offer broad protection against OROV and its reassortment to effectively confront future outbreaks.

OROV has the potential to emerge as a notable public health threat owing to its broad host and vector range, diverse environmental distribution, capacity for severe disease, and the presence of human-infecting reassortants that potentially represent different serotypes. Despite the expanding research on OROV compared with that on other arboviruses, our understanding of this virus remains relatively underdeveloped in the Americas. Strengthening international collaboration, particularly among Latin American countries where OROV is endemic, and increasing funding to address these knowledge gaps are crucial. Such efforts will advance the understanding of OROV, shed light on other neglected diseases, and inform public health strategies. Thus, OROV is a prototypical neglected tropical disease that requires urgent investigation to assess and mitigate its potential health impact.

## Supporting information

**S1 Dataset.  Database of SatScan cluster analyses on confirmed ORO cases from 2024, based on data from the Brazilian Ministry of Health.** It includes case locations along with climatic, environmental, and socioeconomic information for each municipality.
(XLSX)

**S1 Fig.  Map of Brazil showing the number of ORO cases (2024) and changes in pasture cover (2020–2023) across municipalities within high and low SatScan clusters.** The map was built using ArcGIS version 9.1. Source of map base layers: IBGE: https://www.ibge.gov.br/geociencias/cartas-e-mapas/mapas-estaduais.html. Open-source CC BY 4.0 license.
(PNG)

## Author contributions

**Conceptualization:** Camila Lorenz, Thiago Salomão de Azevedo, Francisco Chiaravalloti-Neto.

**Data curation:** Camila Lorenz, Thiago Salomão de Azevedo.

**Formal analysis:** Camila Lorenz, Thiago Salomão de Azevedo.

**Funding acquisition:** Camila Lorenz.

**Investigation:** Camila Lorenz, Thiago Salomão de Azevedo, Maria Anice Mureb Sallum, Francisco Chiaravalloti-Neto.

**Methodology:** Thiago Salomão de Azevedo.

**Project administration:** Maria Anice Mureb Sallum, Francisco Chiaravalloti-Neto.

**Software:** Thiago Salomão de Azevedo.

**Supervision:** Maria Anice Mureb Sallum, Francisco Chiaravalloti-Neto.

**Visualization:** Camila Lorenz, Thiago Salomão de Azevedo, Maria Anice Mureb Sallum, Francisco Chiaravalloti-Neto.

**Writing – original draft:** Camila Lorenz.

**Writing – review & editing:** Thiago Salomão de Azevedo, Maria Anice Mureb Sallum, Francisco Chiaravalloti-Neto.

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
