## [Decision Letter · Decision Letter 0]

PONE-D-25-07911Oropouche fever outbreak in Brazil: key factors behind the largest epidemic in historyPLOS ONE

Dear Dr. Lorenz,

Thank you for submitting your manuscript to PLOS ONE. After careful consideration, we feel that it has merit but does not fully meet PLOS ONE’s publication criteria as it currently stands. Therefore, we invite you to submit a revised version of the manuscript that addresses the points raised during the review process.

The reviewers recommend "Major revision" for the manuscript. The authors need to clarify information about virus, climatic data and socio-economical data

We look forward to receiving your revised manuscript.

Kind regards,

Victoria Pando-Robles, Ph.D.

Academic Editor

PLOS ONE

3. We note that Figures 1, 2 and 3 in your submission contain [map/satellite] images which may be copyrighted. All PLOS content is published under the Creative Commons Attribution License (CC BY 4.0), which means that the manuscript, images, and Supporting Information files will be freely available online, and any third party is permitted to access, download, copy, distribute, and use these materials in any way, even commercially, with proper attribution. For these reasons, we cannot publish previously copyrighted maps or satellite images created using proprietary data, such as Google software (Google Maps, Street View, and Earth). For more information, see our copyright guidelines: http://journals.plos.org/plosone/s/licenses-and-copyright.

a. You may seek permission from the original copyright holder of Figures 1, 2 and 3 to publish the content specifically under the CC BY 4.0 license. 

Reviewers' comments:

Reviewer's Responses to Questions

**Comments to the Author**

1. Is the manuscript technically sound, and do the data support the conclusions?

Reviewer #1: Yes

Reviewer #2: Partly

2. Has the statistical analysis been performed appropriately and rigorously? 

Reviewer #1: Yes

Reviewer #2: Yes

3. Have the authors made all data underlying the findings in their manuscript fully available?

Reviewer #1: Yes

Reviewer #2: No

4. Is the manuscript presented in an intelligible fashion and written in standard English?

Reviewer #1: No

Reviewer #2: Yes

5. Review Comments to the Author

Reviewer #1: The article is very well written and easy to read and follow the authors' reasoning. I believe that it would be important to review the English to improve the reader flow.

The results and discussion do not present the findings related to the socioeconomic factors that appear in the table, and these are the only ones were not included. I suggest add them to the text.

Reviewer #2: The manuscript by Lorenz et al. addresses the study of factors that have contributed to cases of OROV in Brazil, considering various parameters and identifying areas of high and low risk associated with reported cases. The study concludes that temperature and land use are key factors for the re-emergence and high prevalence of cases. This research addresses an issue of global significance, rendering it relevant and worthy of consideration for publication. However, certain deficiencies in the manuscript necessitate the inclusion of precise information for clarification.

The climatic variables have been obtained from WorldClim; however, this platform presents challenges in acquiring climatological data. It is recommended that the authors include excel tables of the most representative years of their analysis as supplementary material, enabling readers to comprehend the values obtained and compared for the analysis.

It is essential to elucidate the data used to train the models. Specifically, the source of records for observations of potential vectors and which vectors were considered in developing prediction models for the geographical distribution of vector species should be clarified. If the ecological niches were derived solely from case data, a database must be integrated into the supplementary material to validate this information. Furthermore, it is suggested that data on potential vectors supporting the distribution and areas of high and low risk be included.

Given the relative unfamiliarity with this virus, it is advisable for the authors to incorporate relevant information about the virus in the introduction. This could include details on its structure, genomic organization, primary reservoirs or vectors, and cellular tropism and replication mechanisms. This information could be presented in a concise manner within a couple of paragraphs in the introduction section. The database provided contains data up to the year 2000. How do the authors address the climate data for the most recent 24 years? How does it correspond to current weather conditions? Furthermore, how was this issue resolved for the analysis in this study?

Minor observations:

Results:

Line 141: "the factors the factors" (repetition noted).

Line 142: Figure 1: Is there a correlation with potential vectors or reservoirs?

Table 1: The areas designated as high and low risk should be accompanied by the corresponding municipality or region, facilitating an estimation of potential expansion of ORO cases outside Brazil. What are the environmental parameters of previous years? Is there a correlation between temperature and previous OROV outbreaks? Is there a relationship between LCLUC in previous years and the presence of the vector? include the exact value of p for each row and the statistical test used.

Discussion:

Line 184: Other potential vectors should be defined and substantiated.

Line 226: "However, a comparison of deforestation rates between 2020 and 2023 reveals a significant expansion of pasturelands and soybean plantations. These changes in land use have the potential to profoundly influence mosquito populations, affecting factors such as oviposition, population density, and host-seeking behavior." This comparison should be visually represented on a map that overlays the incidence of OROV cases.

Additionally, the potential for Culicoides to transmit this virus remains a subject of speculation.

Conclusions:

Line 286: "into the factors driving" It is suggested that this be addressed as the possible factors, given the limited abundance of studies on disturbance and/or incidence in areas with the observed environmental conditions. As with all arboviral diseases, the acquisition of this disease is multifactorial and cannot be attributed solely to climatic conditions. As the authors have previously noted, it is also dependent on vector susceptibility, human population characteristics, and the immunological status of the population.

Figure Legends:

Figure 1: Define the four variables in the text and label them in the figure. In the methods section, enumerate all variables considered.

6. PLOS authors have the option to publish the peer review history of their article (what does this mean? ). If published, this will include your full peer review and any attached files.

**Do you want your identity to be public for this peer review?** For information about this choice, including consent withdrawal, please see our Privacy Policy .

Reviewer #1: **Yes: ** Maysa Pellizzaro

Reviewer #2: No

---

## [Author Response · Author response to Decision Letter 1]

16 May 2025

We sincerely thank the Reviewers for their careful reading of our manuscript and their helpful comments. Below we respond to the comments of each Reviewer in detail, with Reviewer comments in normal-typeface and our responses in bold-typeface. In addition to the Reviewers’ discussion points, we have performed several changes to make our results more robust and to improve the clarity of our study.

Please, note that the pages and sections mentioned in each response are related to the manuscript with changes tracked.

A: Yes, all the text were adjusted to PLOS ONE's style requirements, including those for file naming.

2. Please note that PLOS ONE has specific guidelines on code sharing for submissions in which author-generated code underpins the findings in the manuscript. In these cases, we expect all author-generated code to be made available without restrictions upon publication of the work. Please review our guidelines and ensure that your code is shared in a way that follows best practice and facilitates reproducibility and reuse.

A: No code was used in our analyses, only user-friendly interface commands in the software mentioned in the Materials and Methods: SatScan, Maxent, and ArcGIS.

3. We note that Figures 1, 2 and 3 in your submission contain [map/satellite] images which may be copyrighted. All PLOS content is published under the Creative Commons Attribution License (CC BY 4.0), which means that the manuscript, images, and Supporting Information files will be freely available online, and any third party is permitted to access, download, copy, distribute, and use these materials in any way, even commercially, with proper attribution. For these reasons, we cannot publish previously copyrighted maps or satellite images created using proprietary data, such as Google software (Google Maps, Street View, and Earth). We require you to either (1) present written permission from the copyright holder to publish these figures specifically under the CC BY 4.0 license, or (2) remove the figures from your submission

A: All map layers used in our manuscript are freely available online. We check copyright information on all figures and update the figure caption with source information: “Source of map base layers: IBGE: https://www.ibge.gov.br/geociencias/cartas-e-mapas/mapas-estaduais.html. Open-source CC BY 4.0 license.

Reviewer #1: The article is very well written and easy to read and follows the authors' reasoning. I believe that it would be important to review the English to improve the reader flow.

A: Thank you for your comment, now we revised the English language (see certificate attached).

The results and discussion do not present the findings related to the socioeconomic factors that appear in the table, and these are the only ones were not included. I suggest add them to the text.

A: Ok we agree, now we have added this issue in Discussion section (see 271-286 lines):

“Notably, our analysis revealed that municipalities within high-risk clusters were often characterized by higher poverty rates and limited access to healthcare, with poverty being intrinsically connected to the prevalence of neglected diseases [43]. Furthermore, the Amazon and northeastern regions of Brazil exhibit the highest levels of climate vulnerability [41]. In Brazil, inadequate living conditions and limited access to healthcare services have substantially contributed to the ongoing prevalence of this disease. Several urban areas, particularly low-income neighborhoods, face major challenges in terms of sanitation infrastructure. Although Brazil provides universal health coverage through its public healthcare system, the Sistema Único de Saúde, access remains uneven, especially in rural regions. In the northeastern part of the country, half of the rural population reportedly lives more than 5 km from the nearest healthcare facility, while 60% live over 10 km away [44]. Furthermore, the distribution of medical professionals is heavily skewed toward urban centers, leaving rural areas underserved. These disparities act as barriers to timely medical care, which, in turn, contribute to the rising number of OROV cases in rural communities, as many residents are discouraged from seeking treatment.”

Reviewer #2: The manuscript by Lorenz et al. addresses the study of factors that have contributed to cases of OROV in Brazil, considering various parameters and identifying areas of high and low risk associated with reported cases. The study concludes that temperature and land use are key factors for the re-emergence and high prevalence of cases. This research addresses an issue of global significance, rendering it relevant and worthy of consideration for publication. However, certain deficiencies in the manuscript necessitate the inclusion of precise information for clarification.

The climatic variables have been obtained from WorldClim; however, this platform presents challenges in acquiring climatological data. It is recommended that the authors include excel tables of the most representative years of their analysis as supplementary material, enabling readers to comprehend the values obtained and compared for the analysis.

A: Ok, we included in S1 Dataset the database used, with location of cases and climatic/environmental information. We compared information between 2020 and 2023 years regarding LCLUC.

It is essential to elucidate the data used to train the models. Specifically, the source of records for observations of potential vectors and which vectors were considered in developing prediction models for the geographical distribution of vector species should be clarified. If the ecological niches were derived solely from case data, a database must be integrated into the supplementary material to validate this information. Furthermore, it is suggested that data on potential vectors supporting the distribution and areas of high and low risk be included.

A: Our ecological niche analyses were derived solely from case data, considering only the locations where autochthonous cases have occurred. Unfortunately, there is no information on the distribution of potential OROV vectors throughout Brazil. Now we have included in the S1 Dataset all the data used regarding ORO confirmed cases.

Given the relative unfamiliarity with this virus, it is advisable for the authors to incorporate relevant information about the virus in the introduction. This could include details on its structure, genomic organization, primary reservoirs or vectors, and cellular tropism and replication mechanisms. This information could be presented in a concise manner within a couple of paragraphs in the introduction section. The database provided contains data up to the year 2000. How do the authors address the climate data for the most recent 24 years? How does it correspond to current weather conditions? Furthermore, how was this issue resolved for the analysis in this study?

A: Thank you for your suggestion, now we have added more information about OROV in Introduction (see lines 46-62):

“OROV, a member of the genus Orthobunyavirus, is characterized by a negative-sense, single-stranded RNA structure and a spherical lipid envelope genome [2]. OROV comprises three single-stranded RNA segments and is enveloped by a helical nucleocapsid that encodes its essential components. However, the details of the OROV replication cycle remain unknown. ORO is a syndrome that mimics other vector-borne diseases, such as dengue, Zika, and Mayaro fever, and is associated with common symptoms such as headache, muscle pain, and fever [3,4]. At least four genotypes of ORO have been identified during outbreaks in Trinidad and Tobago, Peru, Panama, and Brazil [5]. OROV persists in a sylvatic-rural cycle with wildlife hosts and arthropod vectors. Although research is ongoing, nonhuman primates, some wild bird species, and pale-throated sloths may act as hosts [6]. OROV transmission from wildlife to humans occurs mainly through the midge Culicoides paraensis. In addition, Culex quinquefasciatus, a medium-sized mosquito common in tropical and subtropical areas globally, may play a role [7,8]. In temperate regions, Culex pipiens may contribute to transmission, with hybrid zones located in the transition zone between the subtropical and temperate regions.”

We compared 2020- and 2023-years using LCLUC data specific for those years (see S1 Dataset). Climate data for the most recent years, 2000 to 2023, was obtained from the National Institute for Space Research (INPE). We included this information in the text (see line 102).

INPE. National Institute for Space Research. 2025. Available at: https://www.gov.br/inpe/pt-br/acesso-a-informacao/dados-abertos Accessed Jan 2025.

Minor observations:

Results:

Line 141: "the factors the factors" (repetition noted).

A: Ok, we corrected.

Line 142: Figure 1: Is there a correlation with potential vectors or reservoirs?

A: Unfortunately, there is no information on the distribution of potential OROV vectors throughout Brazilian territory.

Table 1: The areas designated as high and low risk should be accompanied by the corresponding municipality or region, facilitating an estimation of potential expansion of ORO cases outside Brazil. What are the environmental parameters of previous years? Is there a correlation between temperature and previous OROV outbreaks? Is there a relationship between LCLUC in previous years and the presence of the vector? include the exact value of p for each row and the statistical test used.

A: We included in S1 Dataset the database used, with location (municipality) of autochthonous cases and climatic/environmental information. Unfortunately, prior to 2023, ORO was significantly underreported and often misdiagnosed as other diseases like malaria or dengue, due to the lack of specific diagnostic tools. For this reason, we decided not to include cases before 2020, as although the virus was circulating in the Amazon region, the high level of underreporting could compromise the reliability of our conclusions. In Table 1, we included the exact value of p for each row; we used a t-test to compare the meaning between both groups, assuming equal variances (see lines 135-141).

Discussion:

Line 184: Other potential vectors should be defined and substantiated.

A: Ok, we added more information about this issue (see lines 180-185):

“This spread highlights the growing threat posed to the large immunologically naïve population across the Americas, coupled with the widespread presence of C. paraensis and other potential urban OROV vectors, including C. quinquefasciatus and C. pipiens, extending from the southeastern United States to Uruguay [23]. These ubiquitous house mosquitoes may participate in urban transmission, although vector competency evaluations have yielded mixed results [7,8].”

Line 226: "However, a comparison of deforestation rates between 2020 and 2023 reveals a significant expansion of pasturelands and soybean plantations. These changes in land use have the potential to profoundly influence mosquito populations, affecting factors such as oviposition, population density, and host-seeking behavior." This comparison should be visually represented on a map that overlays the incidence of OROV cases.

A: Ok, we added this map (see S1 Fig).

Additionally, the potential for Culicoides to transmit this virus remains a subject of speculation.

A: Limited data exists regarding the specific vectors associated with recent urban outbreaks, although viral RNA has historically been detected in biting midges, including Cu. paraensis. According to studies of Pinheiro et al. (1981), a single infected Culicoides was capable of OROV transmission. Infection rates in the midges ranged from 54% to 80% and transmission rates from 25% to 83%. Although laboratory transmission of OROV virus by Cx. quinquefasciatus has been demonstrated, the efficiency of transmission is less than that by the midge. The authors conclude that C. paraensis is the primary vector of ORO virus during urban epidemics. We added some information about Culicoides in Discussion section (see lines 185-191):

“Limited data exist regarding specific vectors associated with recent urban outbreaks, although viral RNA has historically been detected in biting midges, including C. paraensis. According to Pinheiro et al. [24], a single infected Culicoides midge can transmit OROV. Moreover, Culicoides midges are a global public health concern, given that they function as known vectors for several other arboviruses, including equine encephalitis and Schmallenberg virus [25]. Approximately 96% of these midge species feed on the blood of humans and wild mammals [25].”

Pinheiro, F. P., Hoch, A. L., Gomes, M. D. L., & Roberts, O. R. (1981). Oropouche virus. IV. Laboratory transmission by Culicoides paraensis.

Zhang, Y., Liu, X., Wu, Z., Feng, S., Lu, K., Zhu, W., ... & Niu, G. (2024). Oropouche virus: A neglected global arboviral threat. Virus research, 341, 199318.

Conclusions:

Line 286: "into the factors driving" It is suggested that this be addressed as the possible factors, given the limited abundance of studies on disturbance and/or incidence in areas with the observed environmental conditions. As with all arboviral diseases, the acquisition of this disease is multifactorial and cannot be attributed solely to climatic conditions. As the authors have previously noted, it is also dependent on vector susceptibility, human population characteristics, and the immunological status of the population.

A: Ok, we rephrased this line in text for the sake of clarity (see lines 313):

“In summary, our findings provide essential insights into the possible factors involved in OROV transmission …”

Figure Legends:

Figure 1: Define the four variables in the text and label them in the figure. In the methods section, enumerate all variables considered.

A: Ok, we correct now (see new version).

---

## [Decision Letter · Decision Letter 1]

Oropouche fever outbreak in Brazil: key factors behind the largest epidemic in history

PONE-D-25-07911R1

Dear Dr. Lorenz,

We’re pleased to inform you that your manuscript has been judged scientifically suitable for publication and will be formally accepted for publication once it meets all outstanding technical requirements.

Kind regards,

Victoria Pando-Robles, Ph.D.

Academic Editor

PLOS ONE

Additional Editor Comments (optional):

Reviewers' comments:

Reviewer's Responses to Questions

**Comments to the Author**

1. If the authors have adequately addressed your comments raised in a previous round of review and you feel that this manuscript is now acceptable for publication, you may indicate that here to bypass the “Comments to the Author” section, enter your conflict of interest statement in the “Confidential to Editor” section, and submit your "Accept" recommendation.

Reviewer #1: All comments have been addressed

Reviewer #2: All comments have been addressed

2. Is the manuscript technically sound, and do the data support the conclusions?

Reviewer #1: Yes

Reviewer #2: Yes

3. Has the statistical analysis been performed appropriately and rigorously? 

Reviewer #1: Yes

Reviewer #2: Yes

4. Have the authors made all data underlying the findings in their manuscript fully available?

Reviewer #1: Yes

Reviewer #2: Yes

5. Is the manuscript presented in an intelligible fashion and written in standard English?

Reviewer #1: Yes

Reviewer #2: Yes

6. Review Comments to the Author

Reviewer #1: (No Response)

Reviewer #2: The authors have included the suggestions and cleared the prior observations so that the article may be considered for publication. Finally, I would like to suggest that supplementary figure 1 be included in the body of the article.

7. PLOS authors have the option to publish the peer review history of their article (what does this mean? ). If published, this will include your full peer review and any attached files.

**Do you want your identity to be public for this peer review?** For information about this choice, including consent withdrawal, please see our Privacy Policy .

Reviewer #1: **Yes: ** Maysa Pellizzaro

Reviewer #2: No

---

## [Editor Report · Acceptance letter]

PONE-D-25-07911R1

PLOS ONE

Dear Dr. Lorenz,

I'm pleased to inform you that your manuscript has been deemed suitable for publication in PLOS ONE. Congratulations! Your manuscript is now being handed over to our production team.

Kind regards,

on behalf of

Dr. Victoria Pando-Robles

Academic Editor

PLOS ONE